# Exploring a Dualism of Human Rationality: Experimental Study of a Cheating Contest Game

**DOI:** 10.3390/ijerph19137675

**Published:** 2022-06-23

**Authors:** Bodo Herzog, Stefanie Schnee

**Affiliations:** 1Economics Department, ESB Business School, Reutlingen University, 72762 Reutlingen, Germany; stefie.schnee@web.de; 2Reutlingen Research Institute (RRI), 72762 Reutlingen, Germany; 3Institute of Finance and Economics (IFE), Reutlingen University, 72762 Reutlingen, Germany

**Keywords:** guessing game, dualism of rationality, optimality, consistency, psychology, behavioral sciences, policy making, decision making

## Abstract

Rational behavior is a standard assumption in science. Indeed, rationality is required for environmental action towards net-zero emissions or public health interventions during the SARS-CoV-2 pandemic. Yet, little is known about the elements of rationality. This paper explores a dualism of rationality comprised of optimality and consistency. By designing a new guessing game, we experimentally uncover and disentangle two building blocks of human rationality: the notions of optimality and consistency. We find evidence that rationality is largely associated to optimality and weakly to consistency. Remarkably, under uncertainty, rationality gradually shifts to a heuristic notion. Our findings provide insights to better understand human decision making.

## 1. Introduction

Since the seminal work by Nagel [1] as well as Grosskopf and Nagel [2] guessing games have been a fundamental part of experimental psychology and economics. The methodology plays a major role in analysing the depth of human reasoning. Indeed, guessing games corroborate the psychology of bounded rationality in decision making [3,4,5].

Our paper intends to study the determinants of rationality or the »optimality–consistency dualism«. We borrow the term from the wave–particle dualism developed in physics almost 100 years ago [6]. Uncovering a dualism in human psychology would be a seminal contribution regarding a better understanding of human decision making in the real-world.

Already, the sociologist Max Weber proposed a somewhat similar notion. He distinguishes two types of rationality, namely instrumental rationality and value rationality [7]. Instrumental rationality refers to the motive of choosing the optimal means [8]. On the contrary, value rationality describes a behavior by which agents choose consistent outcomes. We resume this idea, yet with a different focus.

Our paper experimentally explores whether human rationality consists of optimality and consistency. Optimality is defined as the best response using all available information and frequently updating all information. On the contrary, consistency is the notion of following all publicly available information or rules without incorporating new private signals (for details, see Section 3).

We utilize a newly designed randomized guessing game. Herzog [9] calls this a cheating contest game. The game is based on the beauty contest game, particularly in the first three rounds (for details, see Nagel [1]). Subsequently, we play the cheating contest game, while keeping the new rule under wraps from the instructor and participants by a randomized setup, as is done in medical studies. Consequently, we reveal whether a participant plays optimally or (rule-) consistently.

The rationale of the game is as follows: under the beauty contest game in general, there exists only one Nash equilibrium at zero. This is a subgame perfect Nash equilibria by elimination of dominant strategies, where the *p*-parameter must be in an interval of 0≤p<1. Hence, if the guesses converge to zero in the first three rounds, the players act optimally and rule-consistently according to the beauty contest game. Yet, under the cheating contest game, the players might not follow this pattern. Indeed, the agent might adapt to a heuristic as the optimality notion to win and subsequently does not follow the rule-consistency notion of the beauty contest game. The hidden sequential game setup reveals the dualism of rationality.

The game structure allows us to analyze the dualism because neither the participants nor the instructor expects any game change. Basically, all players suppose they play the beauty contest game over several rounds. If we detect a behavioral adoption in the cheating game towards a winning notion, we conclude that human rationality consists largely of optimality and weakly of some kind of consistency. We examine two hypotheses:1.Agents picking numbers gradually converging to the Nash equilibrium act optimally and rule-consistently (beauty contest game);2.Agents picking numbers that do not convergence to the Nash equilibrium act heuristically optimally but violate rule-consistency (cheating contest game).

In summary: our study does not reject both hypotheses in the relevant game environment. While rule-consistent behavior implies that an agent follows the rule, violation of consistency entails an adoption towards a new (hidden) rule. In general, this adoption corroborates that human rationality consists of both optimality and consistency. Consequently, optimality alone is insufficient to explain rational behavior. Indeed, rationality is equally intertwined with some kind of consistency behavior.

The dualism of rationality has important policy implications. For instance, shaping behavior towards zero emissions requires largely optimal policies. Yet, decision making is bound by human rationality. We discover that the design of a pecuniary incentive scheme, such as a CO_2_ tax, sustains the optimality notion of human behavior. Moreover, we find that under uncertainty, agents swiftly adapt to even hidden optimality notions, while at the same time breaking the consistency notion. If agents do not frequently update information according to the optimality notion, a policy intervention might fail. Consequently, dissemination of information is a key element to empower human rationality and enhance decision making.

We organize the paper as follows. Section 2 contains a literature review on the topic of rationality and guessing games. In Section 3, we introduce the experimental design. The experimental results and discussion are in Section 4. Section 5 concludes the paper.

## 2. Literature

Keynes [10] is the father of the beauty contest game. He described professional investment strategies as a beauty contest in a newspaper in which people have to pick the 6 prettiest women out of 100, of which they think others will find their selection the most beautiful too. Moulin [11] developed the so-called *p*-beauty contest game, which exports the idea of Keynes to science. The first experimental study of the *p*-beauty contest game was done by [1]. Later on, research on the beauty contest game gained popularity [5,12,13,14,15,16].

Overall, the *p*-beauty contest game empirically attacks the rationality assumption. Nagel [1] finds that the depth of rational reasoning is far shorter than the prescription in theory. She corroborates a finite depth of reasoning, mainly first- or second-order beliefs. Yet, the convergence to the Nash equilibrium occurs due to learning theories [12,17,18,19].

Further modifications of guessing games demonstrate that the effect of the group size as well as the role of information have, likewise, an impact on the notion of rationality. Regarding information, [20] finds that “on average, subjects choose as if they significantly underestimate the rationality of their opponents”. In regard to group size, [2] show that single individuals try to behave rationally, yet the group as a whole behaves adaptively. Similarly, work by [21] finds evidence that social learning, especially imitation, accelerates the learning process. Furthermore, [22] argues that the Nash equilibrium is achieved over time. Models by MacLeod or Herzog [23,24] examine how individuals learn.

We follow this literature; however, we combine the game with a newly designed—variant the so-called *p*-cheating contest game, according to [9]. A cheating contest game is a generalization of the beauty contest game [25,26]. So far, the theory has treated people’s rationality as they would hyper-rational supercomputers. Experimental theory, however, corroborates the limits of rationality [1,27]. Our paper intends to uncover two elements of rationality and follows research on rule complexity [28].

## 3. Materials and Methods

The game of a *p*-beauty contest game has *n* participants in general. They simultaneously choose a number $x_{i}$ from a closed interval between 0 and 100. The winner is the person whose chosen number is closest to the mean of all numbers times the *p*-parameter $\left(p*1/n\sum_{i=1}^{n}x_{i}\right)$, where *p* is a predetermined and known parameter, e.g., p=1/2. The winner receives a fixed amount of money whereas the other subjects receive nothing. If there is a tie, the prize is split equally.

In our game, we play at least three consecutive rounds of the *p*-beauty contest game in order to have a control group. Thereafter, in the treatment group, we play four consecutive rounds of the *p*-cheating contest game. In total, we play seven rounds of the game. However, all participants and the instructor are unaware of the game change after round 3 due to the randomized trials. Participants and instructors only have the description for the *p*-beauty contest game (Appendix B).

In principle, the *p*-cheating contest game follows the same rule as the beauty contest. However, in the cheating contest game of round IV to VII, the instructor announces for each round a winning number (‘announced number’) that is automatically manipulated by a hidden algorithm. Thus, neither the participants nor the instructor observe any game change. They expect—as before—the outcome of the *p*-beauty contest game.

In the beauty contest game of rounds I to III, the guesses convergence to the Nash equilibrium at zero by iteration of dominant strategies. In round IV and subsequently, the algorithm creates a predefined pattern of announced numbers. Now, in the light of the cyclical pattern, the winning notion under the cheating contest game is a heuristic. Hence, anchoring the past outcome is a common heuristic and follows the literature [29]. Yet, if agents adopt to a heuristic, they do not follow the rule-consistency notion of the *p*-beauty contest game.

The rationale is as follows: In rounds I to III of the *p*-beauty contest game, the winning strategy is the Nash equilibrium at zero. Thus, agents must act both optimally and rule-consistently at the same time. Starting the *p*-cheating contest game in round IV, the strategy is different. Agents continuing the strategy do not win. Agents adapting are likely to win. Thus, if we exhibit a continuing convergence to zero until round III, yet not after round IV, we cannot reject hypothesis (1) and (2). However, if agents stick to the Nash equilibrium concept until round VII, we must reject hypothesis (2). Thus, we disentangle two notions of rationality: optimality to win the game and consistency to the rule of the game. Indeed, agents always act optimally, yet modify this notion due to the attention to private signals. Hence, we uncover a dualism of rationality.

We conducted the experiment over several years in order to mitigate a selection and time bias. Based on a combination of laboratory experiments, we created synthetic data of sample size N=4200 similar to the ML method in [30]. The sample consists of randomly selected participants without prior knowledge of the beauty contest game. The average age is 24.3 years (Table 1). The sample consists of 77% males and 23% females. On average, participants have 13.9 years professional experience; yet most are students.

We invited large groups to our laboratory at the same time, and half were allocated to the control group and the other half to the treatment group. All participants are placed at separate desks so that communication is impossible (Appendix B). The instructor clarifies the game and answers any question at the beginning. In order to ensure rational thinking, we follow the literature and award the winner either a pecuniary incentive or a gift. During the experimental game we conduct two questionnaires. One questionnaire after round III, and a second questionnaire at the end (Appendix B Figure A4). Yet, we only use specific socio-demographic questions in this study.

## 4. Results

We separate the analysis into two parts. In the first part, we study the structure of the *p*-beauty contest game of rounds I to III. In part two, we study the *p*-cheating contest game of rounds IV and higher. Comparing both games elucidates the dualism of rationality.

### 4.1. Rounds I to III: p-Beauty Contest Game

Figure 1 depicts the scatter plots from each round to the next. Unsurprisingly, the mean in round I is of 12.8 and declines to 10.0 in round II. Thus, participants convergence to the Nash equilibrium at zero, yet do not choose zero immediately. We corroborate Nagel’s ([1] p. 1325) finding of a low degree of reasoning. Nagel called this response the “naive best response”.

In rounds I to II, we observe numbers range mostly between 10 and 20 (Figure 1). From rounds II to III, most choices are below the red line, meaning that participants converge faster to the Nash equilibrium at zero. Note, in round IV, we utilize for the first time the cheating game. Interestingly, one can already observe anomalies (Section 4.2).

In order to study the depth of reasoning, we analyse the distribution via kernel density plots (Appendix A Figure A1). One finds convergence to zero in round I to III. In round III, the majority of choices are below 10 (Figure A1). After round IV one can observe that the peak is increasing away from zero with a greater variance than in previous rounds.

According to Nagel [1], we find almost 90% of the first round choices are within iteration steps zero to three. Thus, we conclude that agents follow the *p*-beauty contest game until round III. Next, we compute the mean convergence ratio over three rounds, *w*, by
(1)$$w_{Mean}=\frac{Mean_{t=1}-Mean_{t=3}}{Mean_{t=1}}.$$

We find mean convergence of $w_{Mean}=0.473$, which almost coincides with Nagel’s [1] finding of 0.46. Our median convergence is 0.6086 and slightly lower in comparison to Nagel (0.75). This implies that mean numbers of agents approximately half from round I to III.

### 4.2. Rounds IV and Higher: p-Cheating Contest Game

The major finding is illustrated in Figure 2, which aggregates the results of all participants. It displays the outcome of the mean guesses over all rounds in red as well as the announced winning numbers by a green dashed line. In addition, we separate in Figure 2 the green area denoting the beauty contest game and the grey area denoting the cheating contest game.

In the beauty contest game, the mean and announced numbers equally convergence to the Nash equilibrium at zero, such as in the standard beauty contest game (Figure 2). In the cheating game, however, the announced numbers follow a predefined algorithmic pattern. In round IV, the algorithm computes the announced winning numbers based on all guesses’ times a predefined *p*-cheating parameter, which is of p=1.2. Similarly, in round V p=0.7, p=1.1 in round VI and p=0.9 in round VII. The cyclical pattern imposes a weak trend to a Nash equilibrium but with up- and down-swings. Thus, the winning numbers display a cyclical downwards-trending heuristic.

At first, we compare the rate of mean guesses. Until round III, the mean guess declined to almost 15 (Appendix A Figure A2). In round IV, the mean declined further to almost 10 (Appendix A Figure A3). This is unsurprising given we play the *p*-beauty contest game until round III.

Note that after round III we conducted the first questionnaire. The dynamics of round IV, despite the first cheating contest game, do not change the notion of average agents. Yet subsequently, we discover a plethora of cyclical patterns (Figure 2). This is surprising because, for the instructor and participants, nothing has changed, only the winning numbers are now differently computed by our hidden algorithm. The dashed green curve represents the “announced” number pattern and the grey area exhibits the cheating contest game. Surprisingly, players directly follow the announced number pattern with a lag period. You can observe this pattern between the dashed green and red curves in Figure 2. Normally, people should follow the rule of the beauty contest game and choose numbers closer to zero. Yet, after round IV, the announced numbers alter the rationality. Participants do not follow the rule-consistency element of rationality anymore. They largely follow an optimality notion or winning strategy (heuristic) without following the rules of the game.

Studying the subsequent rounds of the cheating game, we reveal an adaption of the game reasoning. Table 2 provides an overview. From rounds I to III, 84% of the participants decrease their guesses according to the rationale in the *p*-beauty contest game. However, this number declines significantly to 27% under the cheating contest game. This is a significant change at a level of 1%.

It is noteworthy that in round V, participants increase their guess in order to win (Figure 2). This demonstrates that agents respond to the notion of optimality, yet violate the notion of rule-consistency. Essentially, in the cheating contest game, players follow a naive heuristic which is optimal to win. The convergence path to the Nash equilibrium at zero is barely visible anymore. Consequently, we cannot reject hypothesis (2).

Thus, agents behave optimally in both environments but adapt from rule-consistent behavior under the beauty contest game to heuristic-consistent behavior under the cheating contest game. Consequently, any human rationality is comprised of an optimality and consistency notion.

Finally, we compute a regression in order to evaluate the individual rationality of the the participants (Table 3). We estimate the following regression models, including control variables:(2)$${\mathrm{Round}-\mathrm{III}}_{i}=\alpha +\beta_{1}*GPA_{i}+\beta_{2}*Prof-years_{i}+\beta_{3}*Age_{i}+\beta_{4}*Gender_{i}+\epsilon .$$

In addition, we regress the model to the difference of round I to III:(3)$$\Delta \mathrm{Round}\;{I-\mathrm{III}}_{i}=\alpha +\beta_{1}*GPA_{i}+\beta_{2}*Prof-years_{i}+\beta_{3}*Age_{i}+\beta_{4}*Gender_{i}+\epsilon .$$

The regression results are denoted in Table 3.

The regression output follows the expected relationship of human rationality. Agents choose numbers significantly closer to the Nash equilibrium: the better the GPA grade, the longer the professional experience and the younger the agent. Interestingly, females chose numbers closer to the Nash equilibrium than males.

The estimation of round I to III adjustment according to Equation (Equation 3) follows our expectations too. Agents with longer professional experience converge faster to the equilibrium. Female, elderly and inferior-GPA agents act more erratically and less gradually from round to round.

Indeed, the behavior in the cheating contest game is different (Table A1 Appendix C). The explanatory power of the regression declines significantly according to R-square. In addition, our findings of the adoption towards a heuristic is supported by the literature in psychology, particularly under uncertainty [29,31]. It is striking that participants are not adhering to a declining pattern according to the Nash concept, although the rule (information) is always the beauty contest game. That demonstrates that participants modify their behavior by hidden private signals, even if they infringe the objective rule of the game. One major limitation that might affect our results is the timing of the first questionnaire after round III. In that regard, we cannot preclude that the first questionnaire has an influence on subsequent rounds. This observation is up to further research.

## 5. Conclusions

Our experiment uncovers a dualism of rationality. The randomized game exhibits a new vantage point about human rationality. To our knowledge, this is one of the first research studies utilizing the idea of a cheating contest game. We postulate and exhibit that rationality is composed of optimality and consistency. Consequently, we uncover a dualism of rationality.

Indeed, we find that participants adapt their behavior after round III and follow a new hidden heuristic. Indeed, a heuristic is optimal to win after round IV but it violates rule-consistency (i.e., beauty contest game). Second, the longer the cheating game is played, the greater the deviation from the beauty contest game and the greater the uncertainty. As a consequence, uncertainty leads to a loss of confidence in the objective rule of the game. Subsequently, agents put less attention to the rule-consistency notion. Instead, the optimality notion is persistent and gains in importance, particularly under uncertainty. Third, our finding corroborates learning theory, however, from a different experimental vantage point. From round to round, agents are learning from private signals, even so embedded by a hidden game structure.

The dualism of rationality has an important policy conclusion, including for environmental and public health research. Shaping the behavior of agents, for instance, to reduce greenhouse gases largely requires an incentive scheme to encourage them to act optimally. A consistent policy notion, such as the rule of law, affects rational humans to a lesser extent, particularly under uncertainty. Indeed, we discover that the optimality notion of rationality is dominant and persistent. Yet, human rationality is composed of both optimality and consistency.

## Figures and Tables

**Figure 1 ijerph-19-07675-f001:**
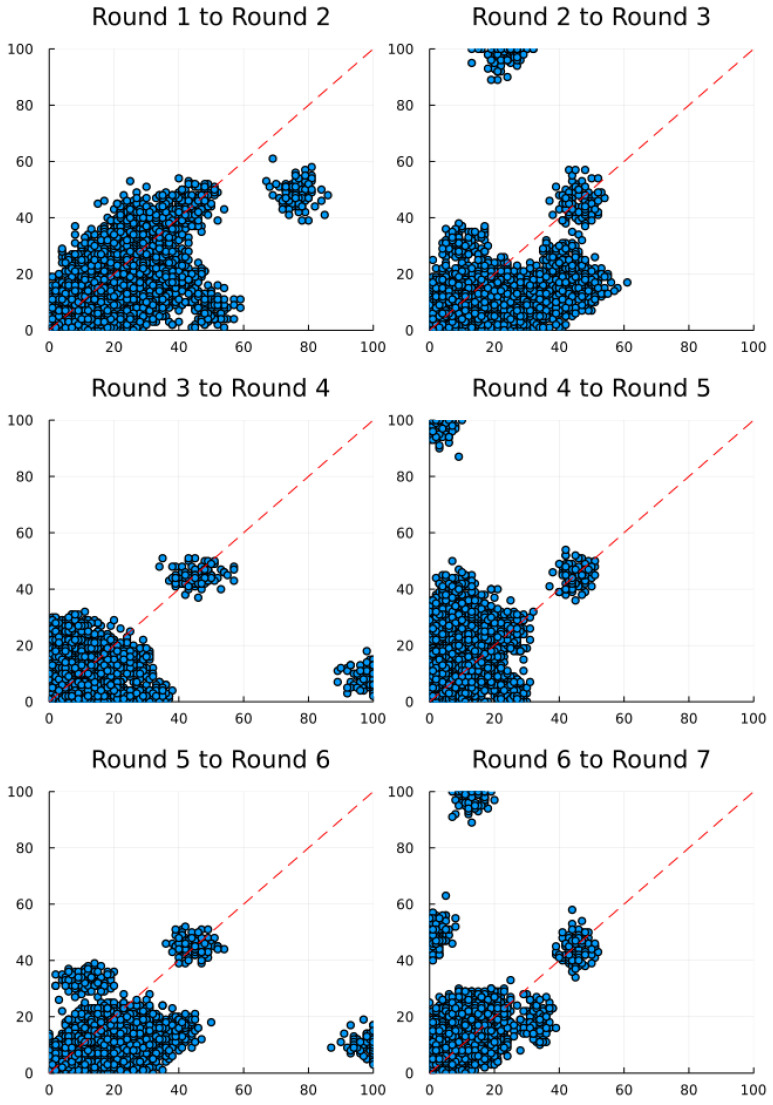
Individual game results. The blue dots denote the responses of all agents in our guessing game.

**Figure 2 ijerph-19-07675-f002:**
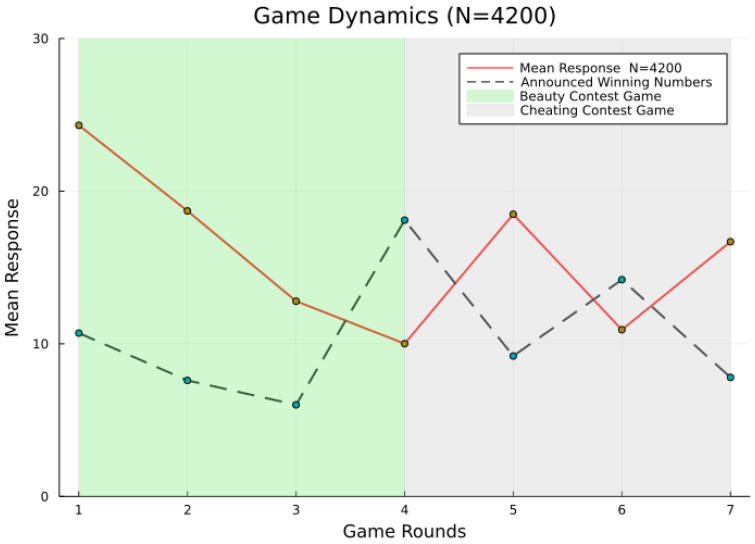
Output of experimental game.

**Table 1 ijerph-19-07675-t001:** Descriptive Statistics.

Variable Names	Mean	Min	Median	Max
Round I	24.3102	0	23.0	86
Round II	18.7029	0	16.0	61
Round III	12.7900	0	9.0	100
Round IV	10.0060	0	8.0	51
Round V	18.4857	0	15.0	100
Round VI	10.9200	0	9.0	52
Round VII	16.6862	0	29.0	94
Age	24.3102	10	23.0	86
Prof. years	15.3055	0	7.0	57
Gender (dummy)	0.5652	0	1	1

The gender dummy variable is one for male and zero for female.

**Table 2 ijerph-19-07675-t002:** Change of agents’ choices (N=4200).

Variables	Round I to III	Round IV to VII
Choices increase	14.26	69.00
Choices unchange	1.78	3.57
Choices decrease	83.95	27.42

Source: authors.

**Table 3 ijerph-19-07675-t003:** Regression Output.

	Round III				ΔRound I to III			
	Model 1	Model 2	Model 3	Model 4	Model 5	Model 6	Model 7	Model 8
(Intercept)	6.917 ***	8.360 ***	6.582 ***	5.259 ***	−6.917 ***	−8.360 ***	−6.582 ***	−5.259 ***
	(0.484)	(0.541)	(0.611)	(0.724)	(0.484)	(0.541)	(0.611)	(0.724)
GPA	0.242 ***	0.251 ***	0.253 ***	0.251 ***	0.758 ***	0.749 ***	0.747 ***	0.749 ***
	(0.017)	(0.017)	(0.017)	(0.017)	(0.017)	(0.017)	(0.017)	(0.017)
Profes. years		−1.985 ***	−1.866 ***	−1.839 ***		1.985 ***	1.866 ***	1.839 ***
		(0.337)	(0.337)	(0.336)		(0.337)	(0.337)	(0.336)
Age			2.096 ***	1.995 ***			−2.096 ***	−1.995 ***
			(0.339)	(0.340)			(0.339)	(0.340)
Gender				0.760 ***				−0.760 ***
				(0.224)				(0.224)
Estimator	OLS	OLS	OLS	OLS	OLS	OLS	OLS	OLS
*N*	4200	4200	4200	4200	4200	4200	4200	4200
$R^{2}$	0.044	0.052	0.061	0.063	0.314	0.319	0.325	0.327

Note: significance of 0.1% = ***. Source: author.

## Data Availability

Data can be found in the paper and appendix and upon request from authors.

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
