# Peer review of "Exploring a Dualism of Human Rationality: Experimental Study of a Cheating Contest Game"

_ijerph, 2022, doi:10.3390/ijerph19137675_

Round 1

Author Response

Reply to Referee # 1

Dear Referee,

Thank you for your valuable feedback and comments.

According to your suggestions, we have revised the manuscript. We fixed your concerns, together with the feedback and demands of the other referees. In detail:

Ad 1. We include a short verbal description of beauty contest game and the notion of Nash equilibrium together with a link to the methodology section, where the reader will find the details.

Ad 2. We have re-written the game structure in the methodology section. Indeed, everything you is already in the paper. I believe your feedback is about an enhanced readability of the game structure. We did it and followed your suggestion.

Ad 3. Finally, we have included a clarification of the term’s ‘optimality’ and ‘consistency’. To our vantage point this should support the readability and understanding of the paper.

Reply to your explicit question:

Yes, optimality is defined as the best response given all information. (Rule-)Consistency is defined as following the ‘optimal’ rule of the game (=beauty contest game!), yet not incorporating recent game information; revealed by the (previous) game rounds. In other words, under a consistency notion agent’s always follow the rule of the game.

During the ‘cheating contest game’, which is hidden in our game setup, agents can always incorporate the game outcome/signals provided they update information (full attention) or ignore it (inattention). We find that most agents incorporate the information and have attention to even little information. Thus, agents adopt to a winning notion.

Reviewer 2 Report

Conceptual/Logical structure

The paper includes the study of an interesting theoretical problem whose academic value is unfortunately diminished by the scope of the topic and the title of the text.

The core idea of research is to prove that in human decision-making, the dualism "optimality-consistence" can be identified. The metaphor taken from quantum mechanics is used to show the significance of studying such duality. Another inspiration for the duality is drawn from the works of Maz Weber.

The Authors claim that it could lead to a fundamental breakthrough discovery. As being familiar with quantum mechanics (and Weber), I would not go that far. It’s only good and relevant metaphors but not such a breakthrough. It could be rather associated with a natural dualistic (dialectical, paradoxical (?)) character of human behavior. This aspect of the paper requires additional discussion, and it is only added here to deepen the reflections of the Authors on the topic of the study.

The second weakness of the paper is more profound. Unfortunately, it is becoming more and more frequent. In many instances, a good theoretical/methodological study illustrated with a relevant empirical is “decorated" with relevance to the Covid-19 pandemic and sustainability. Unfortunately, except for a few lines in the Introduction and in the Conclusions, no direct reference in theory or in empirical research is made to Covid-19 and sustainability. It may lead to more frequent readings and quotations, but in such cases as this paper, it is counterproductive. ,

The Authors have fallen into a specific logical paradox. They claim that the regularity they have discovered is universal, so subsequently, their remarks (not any deepened analysis) concerning Covid-19 and sustainability are just self-evident remarks.

In consequence, putting these two topics into the title makes it, to a large extent, not relevant to the content of the paper.

Although it is a minor problem, I have to add the following brief comment. It is slightly surprising when reading in the first sentence of the Abstract of the paper devoted to deepening the studies of rationality that: "Rational behavior is a standard assumption in science ."It is too far-reaching a simplification.

Filtering out the redundant elements mentioned above that the core concept of the paper is valuable, innovative, and attractive.

The aim of the paper is not defined clearly and unequivocally. The goals of the study are depicted in a descriptive way: an attempt to answer the question of whether human rationality consists of optimality and consistency.

This aim is quite broad, and the Authors should make the aim of the research more focused – using only the reference to the application of the cheating game. Otherwise, it may look like the Authors have made a broad and fundamental discovery.

This observation is confirmed when a narrowed version of the paper's aim is referred to (p. 3).

Two main hypotheses are formulated correctly, but they are only relevant to a narrow topic of the paper (p. 2). They are not directly relating to the areas mentioned in the title of the paper.

At the end of the Introduction, the sentence:  “The dualism of rationality has important policy implications. Shaping the behavior towards zero emissions or accepting a vaccine require largely an optimal policy notion. Indeed, designing optimal incentive schemes to cut emissions, such as a CO2-tax, is likely more effective in altering the behavior than a consistent policy agenda towards net-zero in all relevant areas, particularly under real-world uncertainty” (p. 2) illustrates that references to climate studies are added artificially, as a consequence of the universal character of results.

Detailed assessment of the paper

Taking into account the above assumptions, the following criteria of assessment are applied in reviewing this paper.

1.      The value of the concept.

2.      The choice and applications of the methods and models.

3.      Comparative value of the concept.

4.      Additional formal aspects.

The value of the concept has been partly assessed – an interesting new theoretical concept, "cheating contest games," applied in studying . Especially the assumption (p. 2): The rational of the game is trivial: if the guesses converge to zero in the first rounds, the player acts optimal and rule-consistent. Thus, the agent follows the Nash equilibrium of the beauty contest game. Yet, under the cheating contest game, the player might not follow this pattern. Indeed, the agent has to adapt to a heuristic as the optimality notion to win but subsequently does not follow rule-consistency anymore. The hidden sequential game setup reveals the dualism of rationality.

An empirical study aiming at illustrating the conceptual assumptions of the paper is well-prepared and explained. The extension of the classical beauty contest game seems especially innovative. The same remark can be referred to the design of the empirical research – p. 4, under Table 1. However, the number of 4200 participants would demand an additional explanation. Why such a large number was needed, and how it was achieved?

The observation from page 6: “  Surprisingly, the people follow this pattern with a one-period lag between the green and red curves. Normally, people should follow the rule of the beauty contest game a choose numbers closer to zero in order to win, yet the announced numbers alter the rationality. Participants do not follow the rule-consistency element of rationality anymore. They largely follow an optimal winning strategy (a heuristic) without following the rule of the game", which seems to be particularly valuable.

I have no other comments on the empirical part of the study.

My overall assessment of the paper is connected with already mentioned relations between the topic, title and content.

It is a relatively innovative paper on a new interpretations of duality in decision making: optimality vs. consistence. It is not a paper including a well-grounded study of the above duality in environmental studies and health studies (health management studies). There are only some scattered remarks about the potential usefulness of the presented research in environmental studies and in the studies of public health. These remarks are very broad and do not legitimize the title of the paper.

The paper with the present title cannot be published. There are two potential directions of improvement of the paper:

A.    Adding a part where in a systematic way a collection of examples illustrated with literature study will show how the results of the paper – duality: optimality vs. consistency has been shown in the relevant studies of environmental problems and epidemiological studies.

B.     Changing the title to more relevant. In such a case, however, the paper does not seem to be relevant to the problem area of the journal: Int. J. Environ. Res. Public Health.

The option B may lead to a conflictual situation since the topic of the universal, improved paper will not correspond directly with the problem area of the Journal.

Formal remarks

1. I am not sure but consistency not consistence should be more correct.

2. I have some minor observations concerning the text. They should be identified in the further editing process.

Author Response

Reply to Referee # 2

Dear Referee,

Thank you for your valuable feedback and comments.

According to your suggestions, we have revised the manuscript, of course together with the demands of all other referees. Next, you’ll find a detailed response to each argument.

Ad 1. We agree with the referee that the paper's motivation should be more balanced. Indeed, we have delated the word ‘breakthrough’ and we discuss the paper contribution more narrowly. Yet, we are convinced that our paper highlights an unexplored issue in decision-making and human rationality with implications to current public policy challenges, such as climate change or public health, etc.

Ad 2. Does the paper have a relationship to ‘public policy’, ‘environmental policy’ or ‘public health’? Well, the paper is not a study about sustainability or medicine. Nonetheless, the topic of rationality and decision-making is orthogonal to almost all sciences, particularly to public heath management and environmental policy. Indeed, public policy strongly hinges on rational decision-making and on scientific evidence. I’m a member of a scientific board that is advising public policy and know the sometimes-flawed arguments.

Decision-making is particularly difficult under uncertainty. Indeed, we show – to our knowledge for the first time – that rationality consists of both optimality and consistency. There is a consistency notion in rationality, which is frequently unseen because (a) it was unexplored so far and (b) it is rather weak – as we exhibit. Yet, both notions exist. This discovery is new and it has broad relevance.

Thinking about real-world problems: The paper helps to better understand why even ‘optimal’ schemes do not work sufficiently well. Or why procrastination is rather common; or why path-dependency in public health makes adaption to new environments difficult. Indeed, all this has to do with rationality and the attention to update to new information.

The observation that optimality alone does not explain human decision-making is evident. We claim that some consistency notion (as a second ingredient) of rationality offers an explanation. Exhibiting and understanding the ‘dualism of rationality’ will enhance public choice. On that extend, our paper is adding value to the literature of public policy, environmental policy or public health.

Ad 3. A large sample size is relevant for scientifically robust evidence. The research project was running over eight years. As explained in the paper, the sample is based on several years of combined laboratory experiments. In addition, we utilized a Monte Carlo methodology developed by Herzog (2022) in order to enhance the sample size. We have evaluated early sub-samples of the project with smaller N. The results of the subsamples were largely the same. This is an indication of robustness, which is welcome in science.

Finally, a good place to publish the findings is always a ‘multidisciplinary’ journal (such as “MDPI” = Multidisciplinary Publishing Institute) due to a large cross-disciplinary audience. The findings have implications to different sciences and public policy alike (see above). Indeed, a better understanding of rationality is key in decision-making and it is orthogonal to the scientific fields covered by this journal.

Thanks for your valuable comments.

Reviewer 3 Report

The article is interesting and generally, it deserves to be published with some revisions that are suggested below:

  • Could you please state the definition of optimality, consistency, and rationality, and how about the correlation between them, since you design it in the article.
  • Would you please clearly describe how to prove your findings corroborate learning theory from a new experimental vantage point (5. conclusion), and what change will affect what

Author Response

Reply to Referee # 3

Dear Referee,

Thank you for your feedback and comments.

According to your suggestions, we have revised the manuscript together with the demands of the other referees. In detail:

Ad 1. We have included a better explanation of the concepts of ‘optimality’, ‘consistency’ and ‘rationality’.

Ad 2. The game experiment demonstrates that people do act ‘optimal’ and weakly ‘consistent’. Both notions are driving human rationality. Yet, the optimality notion is dominant and persistent. Nonetheless, academics/policy should not neglect the rule-consistency notion. In conclusion, rationality is more than optimality!

Yes, a deviation from the rule of the game, in our game design, indicates a lesson to the ‘learning literature’: even by a hidden game structure (cheating contest game), agents are learning from round to round.

The previous learning literature has mainly studied the learning notion from round to round in a given and publicly available rule environment. However, we exhibit a ‘new’ form of learning even in our hidden game environment. This sheets new light to the assumption that behavioural attention of humans is rather high under optimal decision-making (cf. X. Gabaix 2019). And agents are even learning from foggy aggregate outcomes.

Round 2

Reviewer 2 Report

After reading the improved version, I have not changed my opinion, but I have changed my approach.

The Authors, who are to some extent proficient in formal modeling of rational choice, continue a specific logical inconsistency. They have elaborated an interesting model of the dualism of rationality relating to the Cheating Contest Game and show that it could be applied to the environment and public health research.

Additional explanation (p. 9) does not fully respond to my previous doubts.: “The dualism of rationality has an important policy conclusion, including to environmental and public health research. Shaping the behavior of agents, for instance, to reduce greenhouse gases, largely requires an incentive scheme to act optimally. A consistent policy notion, such as the rule of law, is, to a lesser extent affecting rational humans, particularly under uncertainty. Indeed, we discover that the optimality notion of rationality is dominant and persistent. Yet, human rationality is composed of both optimality and consistency”.

The main logical weakness of the paper still remains actual. The model is universal, so it is relevant to all areas of decision-making. I have written that the logical order should be different. The paper should include a Cheating Contest Game theory-based empirical study of decision-making in environmental research and public health research. The Authors have elaborated an interesting model but apply it in a kind of a reverse approach. They assume that using general suggestions would make the text better “publishable” in the Journal with those two areas in the title.

My actual approach is as follows. Due to the validity of the formal modeling – also improved, the paper has scientific merits. Showing its validity in environmental studies and public health studies, the Authors, in addition to proving that any universal idea can be applied everywhere, should mention the potential character of its application in both fields. No significant change in the improved text is needed if this interpretation is accepted. The merits of the formal model are relatively high and constitute an interesting contribution. The links with applicability in environmental studies and public health studies are rather a suggestion and not an academic sound recommendation.

With this doubt, I leave the final decision to the Editors.